# Mechanisms of mistrust: A Bayesian account of misinformation learning

**Lion Schulz**[1]*, **Yannick Streicher**[1], **Eric Schulz**[1,2], **Rahul Bhui**[3,4], **Peter Dayan**[1,5]*

**1** Max Planck Institute for Biological Cybernetics, Tübingen, Germany, **2** Helmholtz Institute for Human-Centered AI, Helmholtz Munich, Munich, Germany, **3** Sloan School of Management, Massachusetts Institute of Technology, Cambridge, Massachusetts, United States of America, **4** Institute for Data, Systems, and Society, Massachusetts Institute of Technology, Cambridge, Massachusetts, United States of America, **5** University of Tübingen, Tübingen, Germany

* lion.schulz@tue.mpg.de (LS); dayan@tue.mpg.de (PD)

**Data availability statement:** The code is currently hosted in a private repository on our institute gitlab. We will make this available on gitlab upon acceptance in a dedicated repository.

## Abstract

From the intimate realm of personal interactions to the sprawling arena of political discourse, discerning the trustworthy from the dubious is crucial. Here, we present a novel behavioral task and accompanying Bayesian models that allow us to study key aspects of this learning process in a tightly controlled setting. In our task, participants are confronted with several different types of (mis-)information sources, ranging from ones that lie to ones with biased reporting, and have to learn these attributes under varying degrees of feedback. We formalize inference in this setting as a doubly Bayesian learning process where agents simultaneously learn about the ground truth as well as the qualities of an information source reporting on this ground truth. Our model and detailed analyses reveal how participants can generally follow Bayesian learning dynamics, highlighting a basic human ability to learn about diverse information sources. This learning is also reflected in explicit trust reports about the sources. We additionally show how participants approached the inference problem with priors that held sources to be helpful. Finally, when outside feedback was noisier, participants still learned along Bayesian lines but struggled to pick up on biases in information. Our work pins down computationally the generally impressive human ability to learn the trustworthiness of information sources while revealing minor fault lines when it comes to noisier environments and news sources with a slant.

## Author summary

We are bombarded with information. But how do we learn whom to believe and whom to mistrust? For instance, how do we come to trust one news source's report, while believing that another is biased, produces only useless noise, or might even be lying? And how do we incorporate such possibilities when updating our beliefs? Our work offers a computational and empirical perspective on this learning process. We developed a novel

**Funding:** We gratefully acknowledge financial support from the Max Planck Society (LS, YS, ES, PD), the Fulbright Program (LS), the Humboldt Foundation (PD) and Sloan School of Management (RB). The funders had no role in study design, data collection and analysis, decision to publish, or preparation of the manuscript. The authors were salaried employees of the Max-Planck-Society (LS, YS, ES, PD), and of the Sloan School of Management (RB).

**Competing interests:** The authors have declared that no competing interests exist.

and well-controlled task that allows us to characterize human learning about a host of information sources. We show how people can sometimes be remarkably able to discern lying and helpful sources, even when receiving only uncertain outside feedback. We also show how participants need clear feedback to learn about a news provider's slant.

## 1. Introduction

We are luckily not alone in a complex and uncertain world, and can rely on others for information [1–4]. Friends guide us through social life. Media guides us through political life. While these sources can be helpful for learning, adaptive creatures also need to be wary of the information they receive from other agents [5]. This creates a dual learning problem: We not only need to learn about the issue at hand ("Who is the better candidate for mayor?") but also about the trustworthiness of an information source ("How reliable is the local newspaper?").

Formally, this dual learning learning problem can be understood as Bayesian inference operating on two levels. On the lower level, we need to learn about the state of the world, say the mayor's qualifications. On the higher level, we need to learn about how the evidence we receive from an information source, say the local TV station, lets us make inferences about the state of the world. This Bayesian formulation of learning whether to trust an information source is present in many fields, including in formal epistemology [5,6], cognitive science [7,8], economics [9,10], and applied trust and safety research [11,12].

Whether we solve this dual learning problem appropriately is crucial in many walks of life: For example, when we trust the wrong news sources online, we risk being led astray and falling for misinformation [13,14]. At the same time, mistrust in the right experts may leave us uninformed – as in the discourse on climate change [15–17]. At least equally important, psychopathology is tied closely to the way we view what others tell us. High and adaptive "epistemic trust" has been proposed to be one of the most powerful contributors to good mental health [18,19]. Conversely, out-sized credulity can leave us exposed to exploitation [20–23].

Both personal and political life requires us to learn about information sources of diverse shapes: First, even generally helpful sources are often noisy themselves and may differ in their reliability [7,24,25]: For example, news websites may generally strive to report the truth but nevertheless differ in their journalistic standards [26]. While we should optimally still generally believe their evidence, we should learn to weight more the more reliable source [27].

Another group of sources may be considered unhelpful or untrustworthy: Within these sources, we can draw a crucial distinction: Following a classification by Frankfurt [28], "bullshiting" sources merely produces random noise. In contrast, for a source to "lie", it needs to possess some underlying knowledge of the world state. Adaptive agents respond differently to these categories: "Bullshit" can safely be ignored, like trash websites online [29]. "Lies" in turn can have informational content, and thus license more complex inferences [30–32]. For example, we may interact with a source that constantly tries to confuse and reliably points to a wrong option – letting us infer another option is better, at least in a binary world [8,33].

Finally, information sources can also have biases or slant: For example, media outlets might preferentially laud one candidate. Indeed, media bias is common and its prevalence and formation has been studied both empirically and theoretically [9,34–38]. Savvy agents should again take such bias into account: For example, a left-leaning outlet's endorsement of a progressive candidate carries little information. In turn, a Republican newspaper's endorsement of a Democratic candidate should carry more weight.

Empirically, considerable research has investigated how well people are able to make such inferences from information sources, often under the mantle of advice-taking [7,8,33,39–42]. Advice-taking is involved in a wide range of tasks where we might be uncertain about our own believes, from trivia questions [43] to perceptual judgements [7] and moral decisions [44]. The takeaway is that humans are generally able to learn how useful a source is but this ability is also subject to several limitations. However, little work robustly characterizes the individual trial-by-trial learning computations underlying this dual inference and has investigated unifying learning dynamics underlying the diverse type of sources outlined above.

One crucial determinant for whether we are able to learn about a news source is our level of insight into the ground truth itself: When we have no knowledge about a subject, we are fully at the mercy of a source. In contrast, when we are experts ourselves, or the ground truth is directly observable (like the weather [45]), we can fact-check a source on the go, allowing powerful and quick inferences. Both theoretical Bayesian modeling and empirical investigations support this critical role of having a good feedback signal [5–7].

Here, we investigate people's ability to learn about the trustworthiness of information providers (styled as 'news sources') in a novel task and capture this behavior using Bayesian models. We confront our participants and models with a diverse set of news sources, including helpful, entirely random ("bullshit"), as well as lying and biased sources. We administer two versions of this task. In one, participants learn with full feedback; in the second, they only receive partial feedback, necessitating semi-supervised learning. The tightly controlled, albeit artificial, set-up of our task lets us control rather precisely the information participants receive on a trial-by-trial basis and therefore lets us pin down their learning. This setup also precludes the influence of prior knowledge or motivated reasoning that might be associated with real-world sources [15,46,47]. As a result, we can closely track, isolate, and quantify participants doubly Bayesian learning process.

To preview, we show that people are generally able to distinguish between different types of news sources and learn to respond to them adaptively, in partial accordance with Bayesian models. We show how this ability is attenuated when they only receive partial feedback, that participants show a bias towards believing a source is initially helpful, and that they struggle to pick up on biased information sources.

We begin by formalizing the computational problem and describing our paradigm at a high level before offering analyses of the two versions of our task.

## 2. Methods and results

### 2.1. Ethics statement

Participants in our experiments provided typed informed consent in accordance with procedures approved by the Ethics Committee of the Medical Faculty and Medical Clinic at the Eberhard-Karls-University of Tübingen (approval number 734/2019BO1).

### 2.2. Computational problem and paradigm overview

Imagine you need to make a decision between two political options, such as deciding which policy might be better for your country. We can describe this as a signal detection problem where you need to identify a state $s_t$ (which policy is better) on each trial $t$. Crucially, instead of having perfect knowledge about the state, you only receive noisy information about this,

which we here denote by $X_t$. This can, for example, represent a news report. From this information, you can form a probabilistic belief about the candidates, denoted by the posterior $p(s_t|X_t)$, using Bayes rule:

$$p(s_t|X_t) \propto P(X_t|s_t)p(s_t) \tag{1}$$

This simple Bayesian update is straightforward when we know how good the news report is, that is how $X_t$ is produced from $s_t$. As we alluded to, however, we often lack information about exactly this. As a result, we need first to learn how informative the likelihood $P(X_t|s_t)$ is.

We developed a task that captures this learning problem. Participants were told they were citizens of an alien planet that had two types of policies, green and blue ($s_t \in (green, blue)$). On each trial of the paradigm, which policy was actually better was chosen randomly with equal probabilities. Participants saw a piece of information $X_{I,t}$ from an imaginary news station $I$ pertaining to the quality of the policies, along with a complete or partial feedback signal $Y_t$, and then had to choose which one was better, and rate their confidence in this choice (see Fig 1 for an overview). They were incentivized to make the correct choice and report their confidence faithfully. In the task, different news stations were represented via. different abstract symbols, (here, the pattern of white dots in the purple square in Fig 1B). Each news station had an individual pattern to individuate it, making clear to participants when they are interacting with a novel news station. However, the identity of each news station stayed hidden.

The news report came in the form of a "News Station Council" which consisted of a panel of $n = 5$ experts that endorsed either blue or green. Here, we let $X_{I,t}$ denote the number of blue endorsements out of the 5 experts so that $X_{I,t} \in \{0, 1, 2, 3, 4, 5\}$. This number of blue endorsements was distributed according to a binomial distribution so that each expert had an independent probability of endorsing blue or green. Crucially, this probability was a function of the news station and the state:

$$X_{I,t} \sim \begin{cases} B(n, b_I) & \text{if } s_t = blue \\ B(n, 1 - g_I) & \text{if } s_t = green \end{cases} \tag{2}$$

While this is a relatively simple statistical set-up, it allows us to model a host of different information sources. In our experiment, participants were confronted with four distinct sources whose signal distributions we plot in Fig 1C. We note how these sources generally follow the forms of information providers we outlined in the introduction:

- A **helpful** news source with $b_I = g_I = 0.75$. This source produces more blue endorsements when the blue candidate is better and more green endorsements when the green candidate is better.
- A **random** source with $b_I = g_I = 0.5$. This source is entirely uninformative and does not allow any inferences about the state from its signals.
- An **opposite** source $b_I = g_I = 0.25$. While this source is theoretically as informative as the helpful source, it is inversely so, producing more *green* endorsements when *blue* is in fact the better candidate, and vice-versa.
- A **blue-biased** source with $b_I = 0.9$ and $g_I = 0.5$. In essence, this source is excellent at endorsing a good blue policy but useless at identifying a good green policy. As a result, its overall distribution marginalized over states is skewed towards blue.

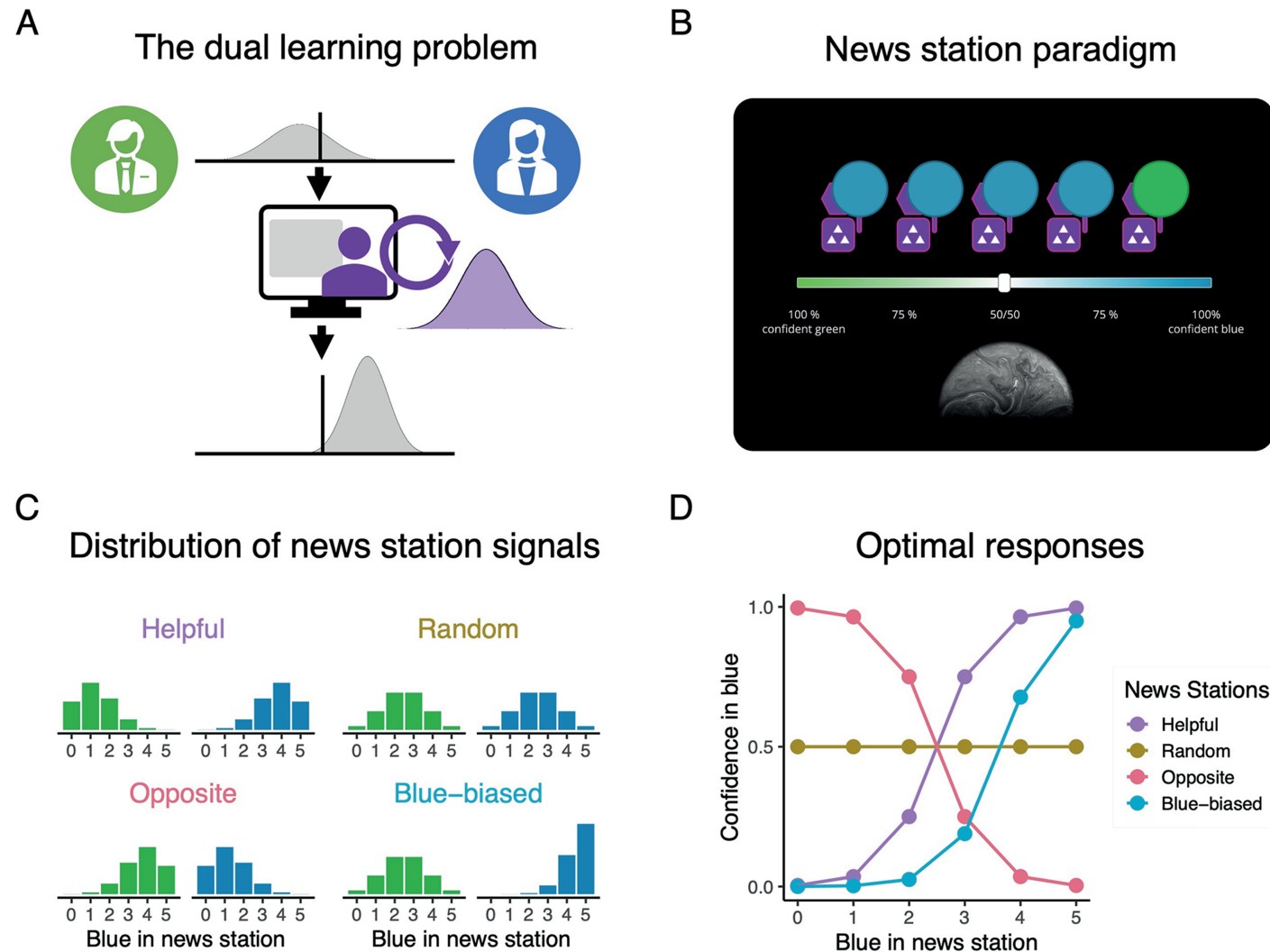

**Fig 1.  Illustration of paradigm.** (A) The dual learning problem underlying how we come to learn about the trustworthiness of a news source. (B) Illustration of the news station task with feedback. Feedback is provided by the planet below the confidence slider lighting up in blue or green after participants have registered their choice. (C) Distributions of blue endorsements in the news stations for the different sources when the ground truth is the green state (in green) and the blue state (in blue). (D) Optimal response curve to the different number of blue endorsements depending on the source.

How should participants optimally react when interacting with such news stations? Fig 1D outlines this, showing the optimal confidence that the blue policy is better, $p(s_t = blue|X_t)$, for all possible signals $X_t$ and sources: For the helpful source, the optimal response is straightforward: the more blue endorsements an agents encounters, the more confident it should be in the blue policy. For the random source, news reports should be entirely ignored, that is, the response should always be 50%—recall that there is no information in "bullshit". For the opposite source, we should optimally invert the response to the helpful source, being more certain that blue is better when we see more green— following Frankfurt, the opposite source represents a simple form of lying. The blue-biased source offers the most complex pattern: While more blue endorsements should still optimally lead to more confidence in the blue candidate, the response curve is shifted: More blue endorsements are necessary to convince

an optimal agent of the blue policy than in the helpful case. Indeed, majority endorsements for blue that are only timid should be met with the belief that the green candidate is better: Imagine a US right-wing website that only sheepishly endorsed a Republican candidate over a Democratic candidate. Crucially, participants were only informed about the general fact that news sources might be more or less informative or biased but were not told about these specific sources.

We can think of the response to a news stations as a psychometric function and can capture this psychometric function via a logistic regression. This regression gives us two parameters that succinctly describe an agent's or participant's responses. These parameters should should differ between sources: The model's slope describes how strongly and in what direction an agent integrates a source's information. It should be positive for the helpful and blue-biased source, meaning the agent increases their confidence in blue being correct the more blue endorsements they see. In turn, the slope should be zero for the random source, meaning that an agent ignores its information, and negative for the opposite source, signifying an inversion of the evidence. The second parameter, the model's intercept, should be equivalent for all but the blue-biased sources, in whose case it should be negative, describing its shifted response criterion. For convenience, we will subtract 2.5 from the number of blue endorsements so that an unbiased response would have an intercept of 0.

## 2.3. Learning and response model overview

How would an optimal agent come to give these responses? Generally, we can extend the Bayesian update from Eq 1 to take into account uncertainty about the parameters $b_I$ and $g_I$ of the sources. In more detail, this means that an agent will have a current belief over these parameter, $p(b_{I,t}, g_{I,t}|\mathcal{H}_t)$, that is based on its learning history, $\mathcal{H}_t \in \{X_{t-1}, Y_{t-1}, \mathcal{H}_{t-1}\}$. To update its beliefs about the underlying state given evidence $X_t$, the agent then marginalizes over this distribution when doing the Bayesian update. In the remainder of this section, we suppress subscript $I$ for readability and suppose that this learning proceeds equivalently, and independently, for all sources. The state update proceeds as follows:

$$p(s_t|X_t, \mathcal{H}_t) \propto \int_{b_t} \int_{g_t} p(b_t, g_t|\mathcal{H}_t)P(X_t|s_t; b_t, g_t)p(s_t) \, db_t \, dg_t \tag{3}$$

Now, to arrive at its belief over the parameters, the second part of the dual learning problem needs to be solved: the higher-level updating. As we mentioned in the introduction, at least some feedback signal or additional knowledge is necessary for this learning to succeed [5–7]. Here, we denote this feedback by $Y_t$. In this paper, we consider two types of feedback: One where the feedback signal is a full revelation of the ground truth on each trial, $Y_t = s_t$, and another where the signal is merely a noisy read-out of the state, being produced by a distribution $\phi$ that is a function of the ground truth, so that $Y_t \sim \phi(s_t)$.

Regardless of feedback type, the optimal learning in both cases also proceeds in a Bayesian fashion. Specifically, the agent combines its prior estimate about the parameters of the source, $p(b_t, g_t|\mathcal{H}_t)$, and current information $X_t$ and $Y_t$ to form an updated belief about the two parameters (again dropping $I$):

$$p(b_{t+1}, g_{t+1}|X_t, Y_t, \mathcal{H}_t) \propto \sum_{s_t} P(X_t|s_t; b_t, g_t)p(b_t, g_t|\mathcal{H}_t)P(Y_t|s_t) \tag{4}$$

This form is equivalent in the full and noisy feedback case but $P(Y_t|s_t)$ takes on different shapes. In the case of full feedback, it merely becomes an indicator $P(Y_t|s_t) \sim \mathbb{1}(s_t = \text{blue})$ so

that $b_I$ only gets updated when the state is blue and $g_I$ only when the state is green. In the case of the noisy feedback, $P(Y_t|s_t)$ is then the conditional probability under the model $\phi(s_t)$ and differently weights the updates of $b_I$ and $g_I$. In brief, in the full feedback case, this update takes the form of a simple and independent closed-form beta-binomial update for $b_I$ and $g_I$ whereas the noisy feedback case requires a more elaborate updating over their joint distribution with no closed-form solution.

To capture participants' expected idiosyncrasies better, we fit this model to their behavior with a total of 9 parameters. These parameters capture a range of phenomena: For example, given the Bayesian set-up of our model, we fit an initial prior to participants, $p(b_{t=0}, g_{t=0})$, letting them be more or less trusting in the early interaction with a news source. We also implement minor biases in the model's learning, for example letting them update quicker towards helpful or unhelpful beliefs, or letting them forget. We note that while these parameters introduce significant degrees of freedom to our model, they nevertheless situate the learning in a Bayesian regime while allowing us to bridge the gap between data and model. We detail both the optimal model computations as well as the parameter fitting in the S1 Text.

## 2.4. Experiment 1: Full feedback

**2.4.1. Experimental details.**   In the first version of our task, participants received full feedback about the true better policy at the end of each trial. In each trial, participants first saw the news station consisting of five experts and had to rate their confidence. They responded using a colored slider from "100% sure green" to "100% sure blue" (via "50/50") (see Fig 1B). After they had rated this confidence, participants received feedback about the ground truth. Participants interacted with each of the sources for a total of 28 trials. We presented the sources in a blocked manner, randomizing the order between participants. Thus, the whole experiment consisted of 4 blocks, each comprising 28 learning trials plus 6 probe trials.

To test participants' beliefs in the sources in a standardized manner, each block ended with an additional six probe trials, clearly demarcated by following a short break, where participants saw all possible constellations of blue and green endorsements $X_{I,t} \in \{0, 1, 2, 3, 4, 5\}$ in random order. Participants did not receive feedback during these trials. At the end of each block, we additionally asked participants how much they thought that the news station improved their decision and how much they trusted the news station. At the end of the experiment, participants filled out a battery of questions relating to trust.

We analyze data from a total of 123 US adult participants collected via Prolific with a broad range of ages and educational backgrounds (see S1 Text for more details).

**2.4.2. Results: Probe trials.**   For a high-level overview of participants' learning success, we first investigated their responses on the final probe trials. Recall that these showed each participant all possible combinations of source endorsements and so let us investigate their full 'revealed' belief in the source.

We plot these responses in Fig 2A showing the average responses across participants. This reveals that participants were able to pick up on the source's attributes (we provide a statistical analysis below): For the helpful source, on average participants were more confident that blue was better the more blue endorsements they saw. In contrast, for the random source, this integration was diminished, and participants tended to relatively ignore its signals. Participants also inverted the evidence that they received from the opposite source, being more confident in *blue* the more *green* endorsements they saw. Finally, we observed a marked shift in the participants' responses to the blue-biased source: When participants only saw a weak

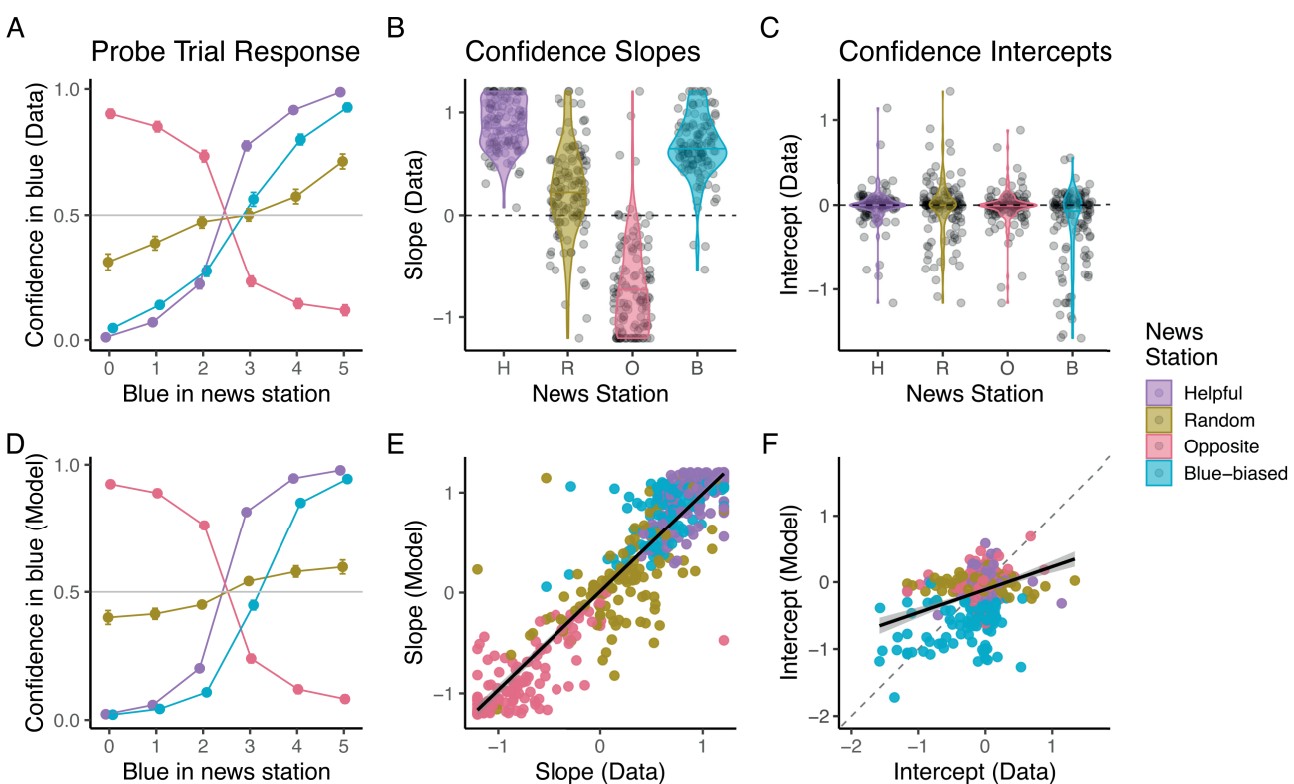

**Fig 2. Probe trial results show good learning convergence and fit of Bayesian model in full feedback task.** (A) Participants' psychometric responses to the sources in the probe trials. Dots represent means and error bars represent standard errors of the mean (partially occluded by the dots). (B, C) Distributions of slopes and intercepts fit to individual participants' probe trial responses. (D) Probe trial predictions of fit model. Dots represent means and error bars represent standard errors of the mean (partially occluded by the dots). (E, F) Correlations of slopes and intercepts fit to individual data and model probe trial responses.

endorsement of blue ($X_I = 3$), their confidence was on average almost indifferent between blue and green.

As we outlined, we can describe this behavior via a logistic regression. We thus fit individual regressions to participant responses (see S4 Text for details). Recall that the regression slope should be positive for the helpful and blue-biased source, zero for the random source, and negative for the opposite source. The second parameter, the model's intercept, should be equivalent for all but the blue-biased sources, in whose case it should be negative. In line with our across-participant analysis, we found that slopes fit to participants' probe trial responses (Fig 2B) were positive for helpful and blue-biased sources, lower for the random sources, and negative for the opposite source. This was supported by a significant main effect of source on slope in an ANOVA ($F(3, 488) = 435.40$, $p < .0001$) and significant differences in Tukey's tests between all sources (all $p's < 0.0001$). Participants' slopes for the helpful, blue-biased, and random source were all significantly higher than zero (one-sample t-tests, all $p's < 0.0001$), indicating that on average participants still held the random source to be slightly helpful. The opposite source's slopes were significantly lower than 0 (one-sample t-test, $t(122) = -17.10$, $p < .0001$), indicating that participants on average inverted its evidence, at least slightly.

In turn, there was no significant difference between the intercepts (Fig 2C) of the helpful, random, and opposite source (Tukey's tests, all $p's > 0.9$) and they were all not significantly different from zero (one-sample t-test, $p's > 0.2$). However, the intercept

of the blue-biased source differed significantly from all the other sources (ANOVA testing main effect of source, $F(3, 488) = 11.82, p < .001$, and Tukey's tests comparing blue-biased means to other means of other sources, all $p's < 0.0001$), and was significantly lower than 0 (one-sample t-test, $t(122) = -5.54, p < .0001$). However, while the intercept distribution was generally shifted negatively for the blue-biased source, we also observed that most participants remained within the ranges of the other three source types.

Participants' responses were well captured by our Bayesian model. This is visible in Fig 2D where we plot the Bayesian model's predictions. Our model was also able to capture individual patterns of responses. To check this, we fit the same logistic regressions to the fit model posterior predictions. We observed strong correlations between the psychometric slopes obtained from the data and model ($r = 0.92, p < 0.0001$). These slopes are the main differentiator between the four sources and thus the main indicator for learning success. We found a weaker albeit still significant correlation between the model and data for the intercept ($r = 0.36, p < 0.0001$). This is unsurprising, given that a considerable number of participants did not, as we discussed, manage to realize there was a bias.

**2.4.3. Results: Learning dynamics.** Our analysis of the probe trials showed that participants were generally able to learn the statistical regularities of these sources - but how did they arrive at these conclusions? To answer this, we next analyzed participants' learning trajectories over the first 28 trials of a block (excluding the probe trials).

First, we again investigated participant averages, tracking their beliefs across the interaction with a source. To do so, we fit logistic regression models akin to the ones described above for each source individually, splitting each block into quarters of seven trials and pooling the responses across participants.

Fig 3A and 3B show the results of this analysis: Already in the first quarter, there are markedly different responses to the four sources, with the blue-biased and random showing lower, albeit positive, slopes than the helpful source. The opposite source in turn shows a slope close to zero (panel A). This distinction develops further across the block with the opposite source's slope becoming negative — that is on average participants began inverting its evidence rather than just ignoring it — and the random source showing a slow decline towards zero — that is participants began to ignore its evidence more. There was a mirrored slight increase in the helpful source's slope, showing a slightly larger integration of its evidence towards the end of a block. In turn, the average intercepts remained equivalent for all but the blue-biased source, whose intercept declined towards the end of the trial. This lowering intercept reflects learning that the blue-biased source was indeed biased. Fig 3C shows the effects of this learning on participants' average accuracies, computed by binarizing their confidence ratings. Initially, participants were able to glean no information from the opposite source, as shown by their random performance in the first block quarter. However, as their interaction with the source went on, their judgement also became more accurate, almost reaching the accuracy of the helpful source by the end of the block. Recall that the opposite source in theory conveys just as much information as the helpful source. This general accuracy pattern was supported by a significant source $\times$ trial interaction in a logistic regression predicting accuracy ($\chi^2(3) = 239.43, p < 0.0001$)

Our model was able to capture these dynamics as is evident in Fig 3D–3F. There, we show the learning dynamics for the posterior predictions of the model. All response patterns are well recovered: Both participants and model are able to develop distinguishable responses to the sources already in the first half, already ignoring the opposite source (Fig 3D). In line with our analyses of the probe trials, we also find that the model is able to pick up quicker (and

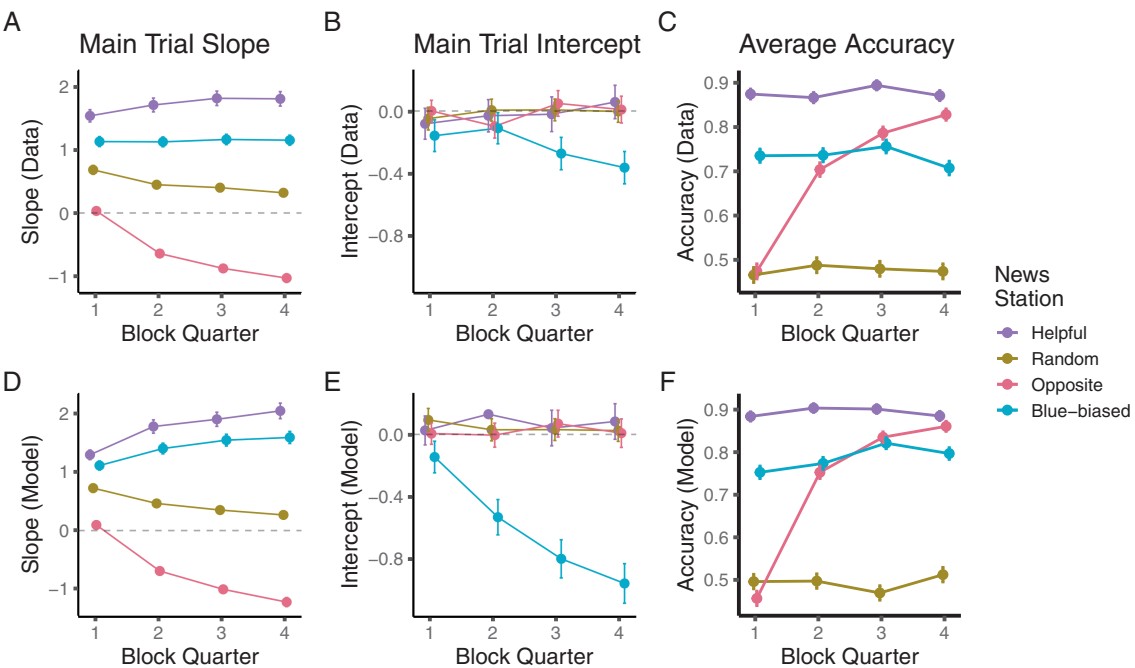

**Fig 3. Pooled regression analysis shows signatures of Bayesian-like model in full feedback version.** (A, B) Slopes and intercepts fit to pooled participant responses, split by block quarter and source. (C) Accuracy split by block quarter and source. (D - F) The same metrics inferred from fit model. Dots in A, B, D, and E represent the mean regression coefficient estimates, and error bars represent their respective standard errors (partially occluded by the dots). C and D show mean accuracies and the error bars standard errors of the mean (again partially occluded).

more strongly) on the bias than the participants (Fig 3E). Accuracy patterns, however, were comparable between the model and data (Fig 3F).

To investigate whether these learning patterns held on an individual level, we next fit logistic regressions individually to the data. To ensure a sufficiently large sample size per regression, we split each participant's 28 trials into two halves. We plot the results of these analyses in Fig 4 with the slope and intercept values in A and B. We observed a significant two-way source × block half interaction for both slope and intercept (slope: $F(3, 976) = 27.17, p < .0001$, intercept: $F(3, 976) = 5.99, p < .0001$).

Crucially, participants' slopes in response to the opposite source were significantly lower in the second half of a block compared to the first (post-hoc Tukey's test, $p < 0.0001$). Participant's intercepts showed a decrease when interacting with the blue-biased source ($p = 0.0005$). Despite trends in the expected directions, there was no significant difference between participants' first and second half slopes when interacting with the opposite ($p = 0.30$) and random source ($p = 0.25$). Slopes also did not differ for the blue-biased source ($p > 0.99$). As we would expect given the lack of bias, intercepts for helpful, random, and opposite sources also did not change between the first and second half ($p's > 0.9$).

Fig 4C shows that participants' individual patterns were also well captured by our fit model: The slopes we recovered for the two halves again showed a high correlation between model predictions and data ($r = 0.90, p < 0.0001$), showing how participants and models similarly picked up on the correct response patterns. As before, this correlation was still significant for the intercept, although to a lesser degree ($r = 0.57, p < 0.0001$), highlighting the difficulty participants had in learning about the bias (Fig 4D).

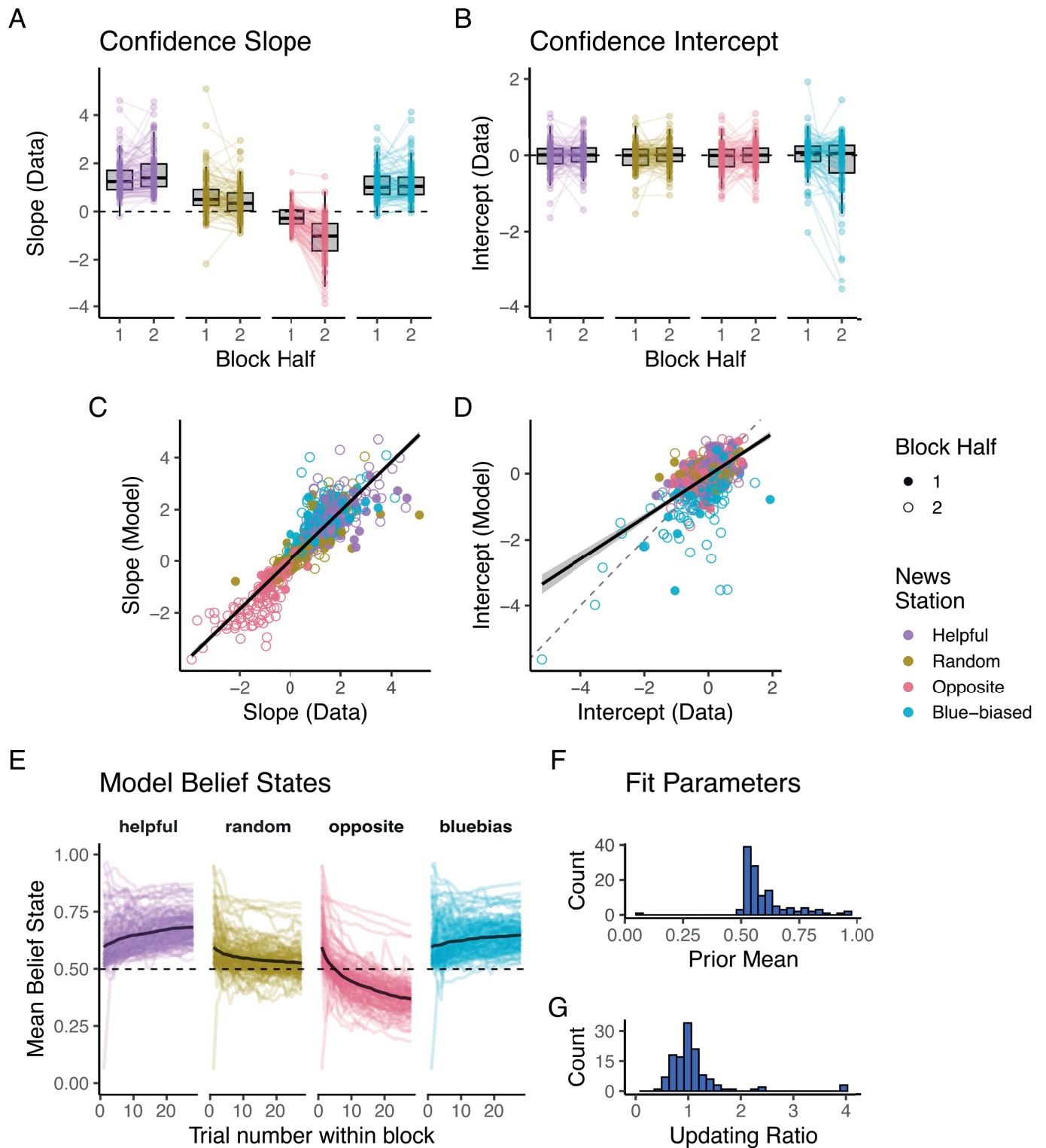

**Fig 4. Individual regression analysis highlights learning and model fit in full feedback version.** (A, B) Slopes and intercept of psychometric curves fit to first and second half of participant responses. (C, D) Correlation between first and second half slopes and intercepts fit to data and fit model. (E) Trajectory of mean over the two belief states. Coloured lines represent individual participants, and black line mean over participants. (F) Distribution of mean over two prior parameters. (G) Distribution of updating ratio.

The model's internal belief states revealed the individual learning process on a more fine-grained level. In Fig 4E, we plot these beliefs, that is, what the model assumes to be the binomial probabilities with which the source generates its information. We plot the mean over the two beliefs (formally, $\mathbb{E}[\mathbb{E}[b_{I,t}]+\mathbb{E}[g_{I,t}]]/2$) to highlight the learning process and note key similarities between this plot and the dynamics we displayed in the pooled regression analysis in Fig 3A and 3D.

We highlight two distributions of fit parameters which lets us quantify aspects of these individual learning dynamics: Fig 4F shows the fit prior belief (averaging the prior for $b_I$ and $g_I$). This affirms our model-agnostic analyses using the slopes: Participants' prior tended to be positively skewed ($\mu = 0.60, \sigma = 0.11$). Additionally, the closed-form belief update within our model let us straightforwardly fit a learning bias, asking whether people tended to learn quicker in favor of believing that a source was helpful (updating ratio > 1) versus that a source was unhelpful (updating ratio < 1). This distribution centered around 1 ($\mu = 1.09, \sigma = 0.56$) indicating that participants did not update in a particularly hopeful or unforgiving manner. This was supported by a one-sample t-test that showed, despite outliers, no significant difference from 1 ($t(122) = 1.80, p = 0.074$).

## 2.5. Experiment 2: Noisy feedback

In the real world, unambiguous feedback is often lacking. In the second experiment, we investigated how well participants were able to cope with noisier feedback when learning how reliable or biased a source was.

**2.5.1. Experimental details.** Participants again interacted with the four different news sources, presented in a randomized blocked manner. However, instead of fully revealing the better policy at the end of each trial, participants had to rely on a noisier source of information, consisting of a panel of experts that were displayed in the same manner as the news station. Fig 5 shows the time course of a trial: Participants again first saw the news station consisting out of five green or blue endorsements, and rated an initial confidence (using the same scale as above). Following this, they were shown the Independent Council, also consisting of

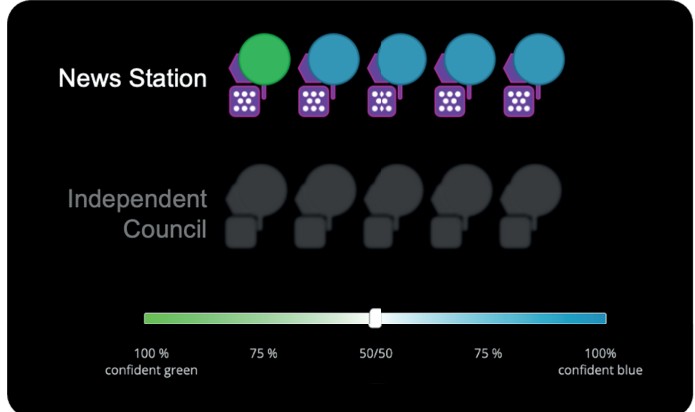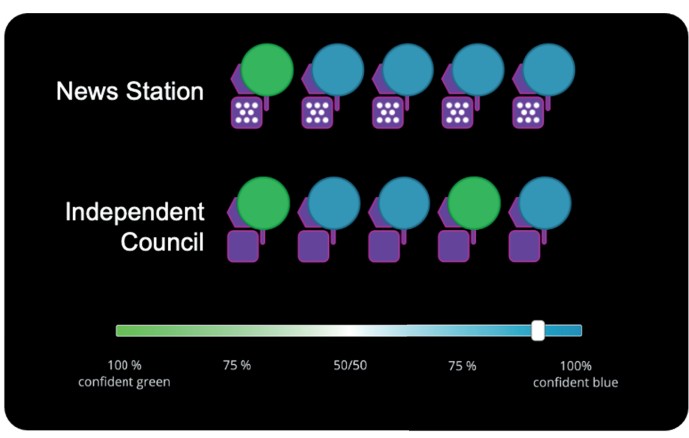

Time

**Fig 5. Illustration of trial in noisy feedback condition.** Participants first saw the news station experts and rated their confidence based on this. They then saw the independent council and were able to adjust their confidence.

five green or blue stochastic endorsements. Based on this, they could revise their confidence, using the same slider.

Participants were told that the "Independent Council" was a helpful, albeit sometimes erring, collection of experts. Specifically, the statistical set-up of this expert panel was such that it was equivalent to the helpful source. Here, we denote the number of blue endorsements as $Y_I$ and the parameter producing this evidence with $b_Y$:

$$Y_{I,t} \sim \begin{cases} B(n, b_Y) & \text{if } s_t = blue \\ B(n, 1 - b_Y) & \text{if } s_t = green \end{cases} \tag{5}$$

Here, we set $b_Y = 0.75$. Participants were extensively informed about this quantity. We conducted analyses to ensure that participants used the independent council's feedback which we report in Fig B in S1 Text in particular.

As before, participants interacted with each source for 28 trials before engaging with 6 probe trials. They also again answered the trust and improvement question at the end of each block and a battery of questions at the end of the experiment.

We analyze data from a total of 111 US adult participants, again collected via Prolific and including a range of ages and educational backgrounds (see S1 Text).

**2.5.2. Results: Probe trials.**   We again first analyzed the probe trials at the end of each block (Fig 6A–6C). This shows how participants displayed individual patterns of responses that were broadly in line with an adaptive response but less so than in the full feedback case: The random source was again associated with a lower slope than the helpful source (Tukey's test: $p < 0.0001$). However, on average, participants still integrated the random source's information, with their average slopes being significantly higher than zero (see Fig 6B, one-sample t-test $t(110) = 11.74, p < .001$). On average, participants managed to invert the opposite source's evidence ($t(110) = -3.63, p < .001$), but this inversion did not mirror in strength the helpful source. While we observed a minute shift in the response to the biased source the individual intercepts, shown in Fig 6, were on the whole not significantly different from zero ($t(110) = -0.39, p = 0.696$).

Most of these patterns were well-captured by the Bayesian model, as is visible in Fig 6D–6F. The model's general response patterns (panel D) matched those of the participants. Again, this was highlighted by the strong correlation between the slopes recovered from fit model predictions and from the data (panel E, $R = 0.80, p < 0.0001$). In line with most participants' inability to realise the blue-biased sources bias, we found only a weak correlation between the model predictions and participant data (panel F, $R = 0.17, p = 0.0003$).

**2.5.3. Results: Learning dynamics.**   How did participants reach these beliefs? To investigate this, we again fit regressions to the main block and investigated their dynamics. Fig 7A and 7B show the results of this analysis. In the main block, participants' response slopes showed less distinction between the helpful, random, and blue-biased sources. Indeed, the slopes in response to the random source were only slightly lower than to the helpful source and only developed this difference over the course of the block. While the opposite source's slope is clearly distinguished from the remaining sources, participants' inverting response is not as strong as before, converging at, on average, ignoring the source rather than inverting its evidence. As to be expected from the probe trials, participants' average intercept in response to the blue-biased source barely shifted away from the remaining sources.

The effects of this slower learning are evident in Fig 7C where we plot the time course of the participants' average accuracies after having seen the news station (but not having seen the helpful independent council): The positive integration of the evidence coming from the

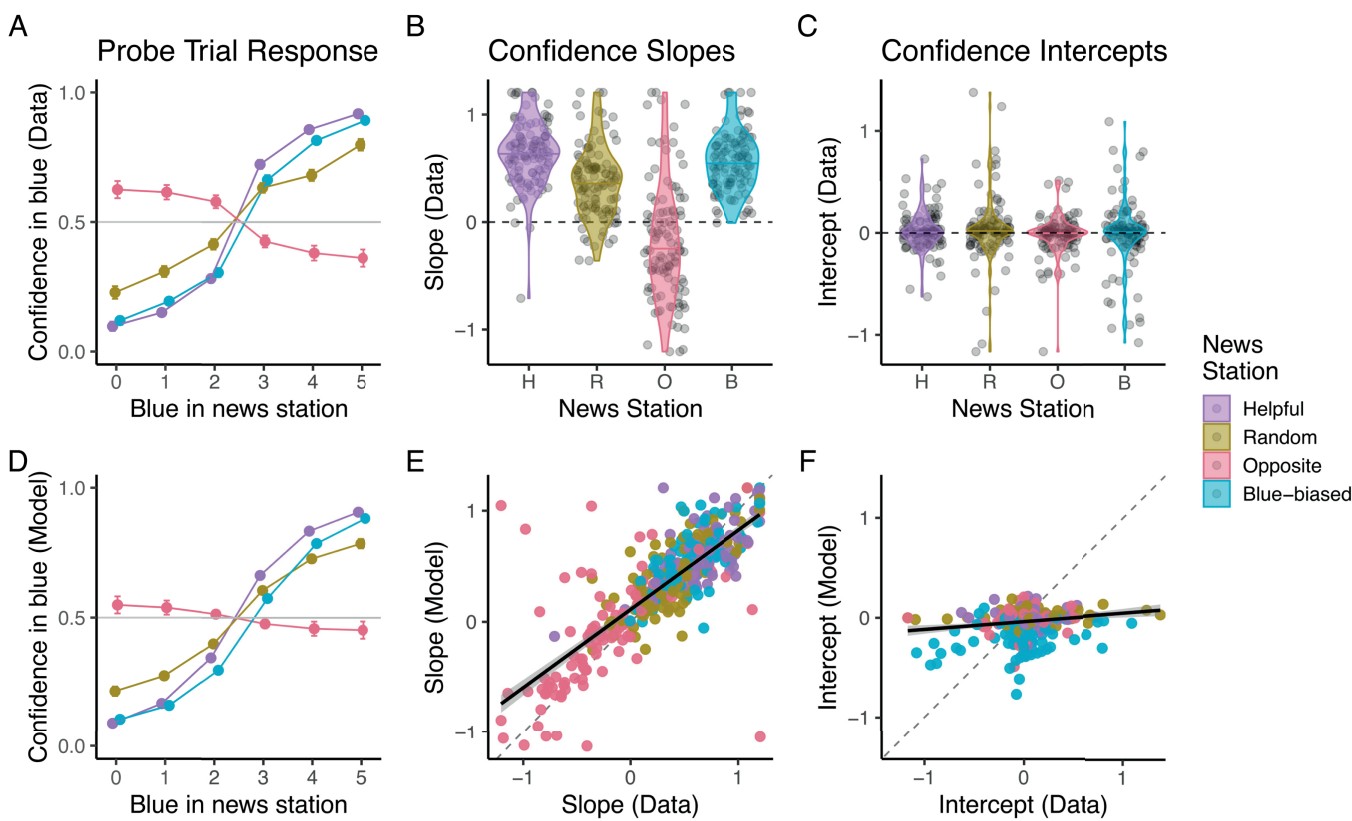

**Fig 6. Probe trials in noisy feedback condition highlight diminished learning convergence and model fit.** (A) Participants' psychometric responses to the sources in the probe trials. Dots represent means and error bars represent standard errors of the mean (partially occluded by the dots). (B, C) Distributions of slopes and intercepts fit to individual participants' probe trial responses. (D) Probe trial predictions of fit model. Dots represent means and error bars represent standard errors of the mean (partially occluded by the dots). (E, F) Correlations of slopes and intercepts fit to individual data and model probe trial responses.

helpful and blue-biased source results in high accuracies throughout the block. Inference from the random source naturally remains at 50 % accuracy, containing no information. In contrast, participants initially fall for the opposite source's misinformation, integrating their information in the incorrect manner: This leads them to perform below chance accuracy in the first half of the block. Only in the second half of the block do they reach accuracy levels that align with the random source, in line with their ignoring of the opposite source's information (panel A). This was again supported by a significant within-block trial number × source effect on accuracy ($\chi^2(3) = 52.52, p < 0.0001$)

Our model was again able to capture these general dynamics as is visible in Fig 7D–7F: The model's response slope only slowly and weakly distinguished the helpful, random, and blue-biased source. The model initially weakly integrated the opposite source and then ignored it towards the end of the block (panel D). There was little distinction in the intercepts (panel E). These learning patterns were reflected in the model's average accuracy (panel F).

Analyses of individual-level responses supported these results, as is shown in Fig 8. We again found a significant block half × source type interaction effect in an ANOVA predicting the slopes ($F(3, 880) = 3.43, p = 0.017$, see panel A). Again, participants' response slope significantly decreased only for the opposite source between the first and second block halves (Tukey's test, $p = 0.027$). All other sources saw no changes in slope ($p's > 0.97$). Conversely,

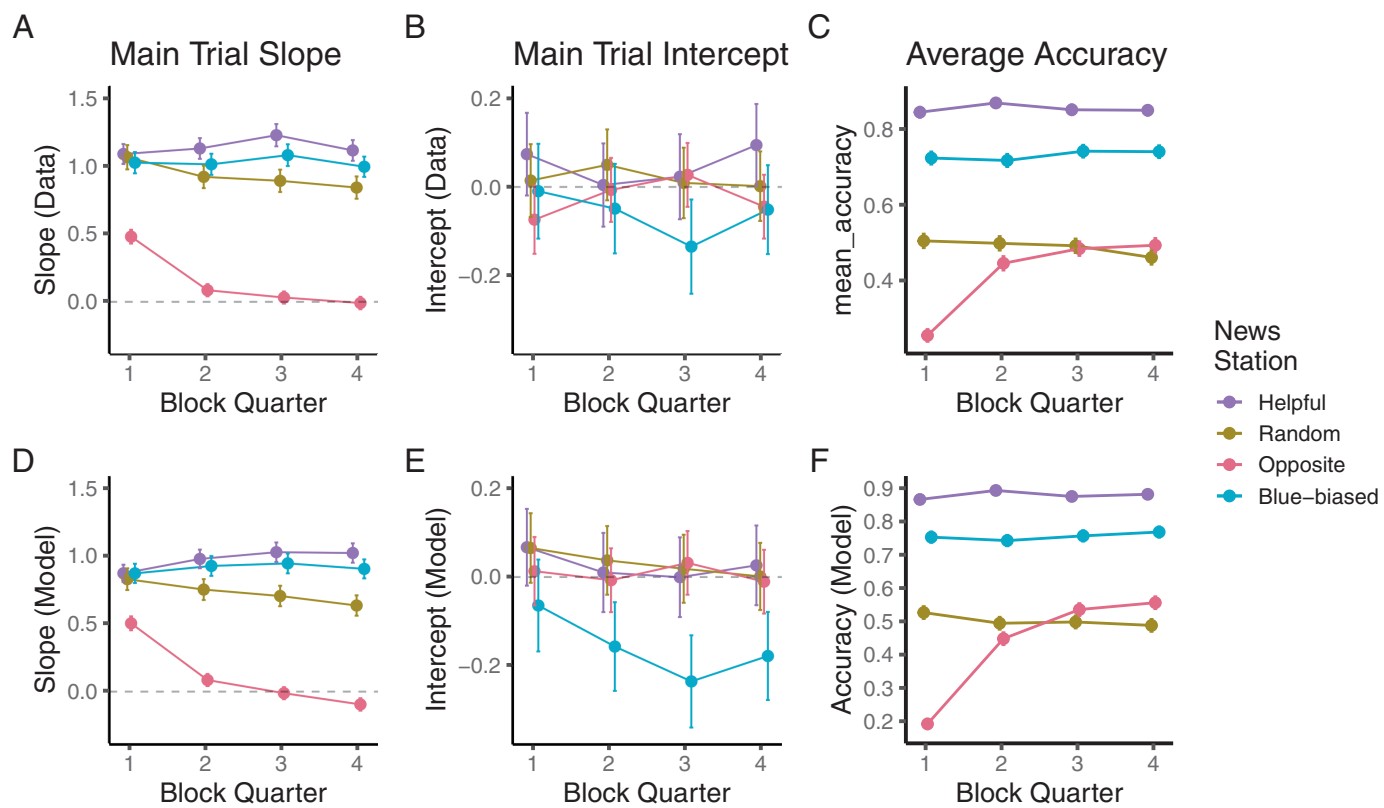

**Fig 7. Pooled regression analysis shows signatures of Bayesian-like model in noisy feedback version.** (A, B) Slopes and intercepts fit to pooled participant responses, split by block quarter and source. (C) Accuracy split by block quarter and source. (D–F) Same metrics inferred from fit model. Dots in A, B, D and E represent the mean regression coefficient estimates, and error bars represent their respective standard errors. Note how these error bars are partially occluded by the dots. C and D show mean accuracies and the error bars depict standard errors of the mean (again partially occluded).

there was no significant shift in intercepts, supported by a non-significant block half × source type interaction effect in an ANOVA ($F(3, 880) = 1.56, p = 0.198$).

Our model captured these individual response patterns well, as is visible in panels C and D: Slopes and intercepts fit to the model's responses correlated with those fit to participants data (slope: $R = 0.88, p < 0.0001$, intercept: $R = 0.59, p < 0.0001$ ).

These individual learning dynamics were also visible in the fit model's belief states which we show in panel E, again pooled across the two beliefs. Learning was mostly evident when interacting with the opposite source, and participants remained largely around their own helpful prior. Indeed, this prior was again skewed positively ($\mu = 0.68, \sigma = 0.11$). We note that due to the non-closed form nature of the belief update, we did not characterize the updating bias as comparable to the full feedback case.

## 2.6. Summary and experiment comparison

In both experiments, the Bayesian model described participants well but not perfectly. How did these conditions compare?

We first compared participants' average accuracies between the two versions: As to be expected, participants showed higher accuracy in the condition with full feedback, and this

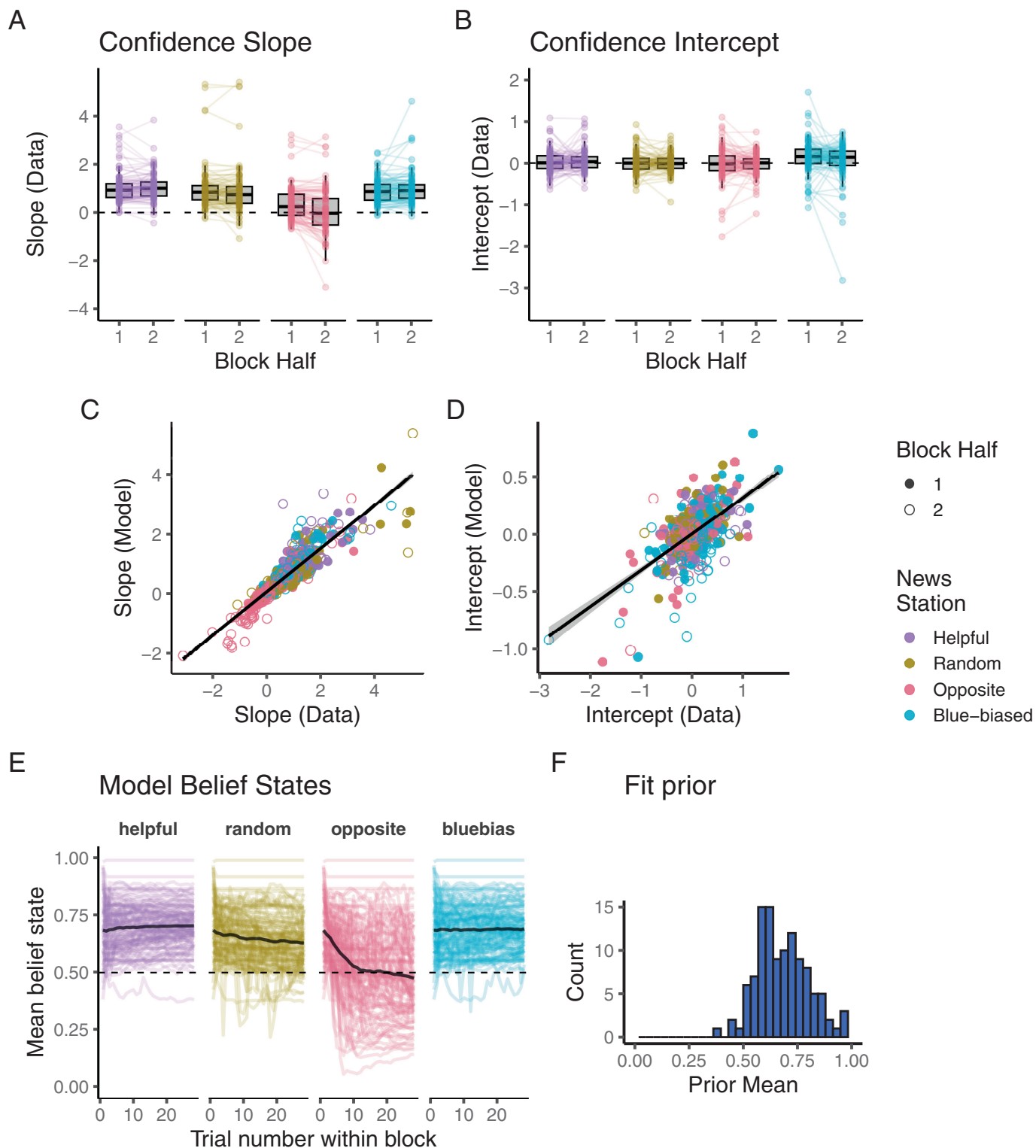

**Fig 8. Individual regression analysis highlights learning and model fit in noisy feedback condition.** (A, B) Slopes and intercept of psychometric curves fit to first and second half of participant responses. (C, D) Correlation between first and second half slopes and intercepts fit to data and fit model. (E) Trajectory of mean over the two belief states. Coloured lines represent individual participants, black line mean over participants. (F) Distribution of mean over two prior parameters.

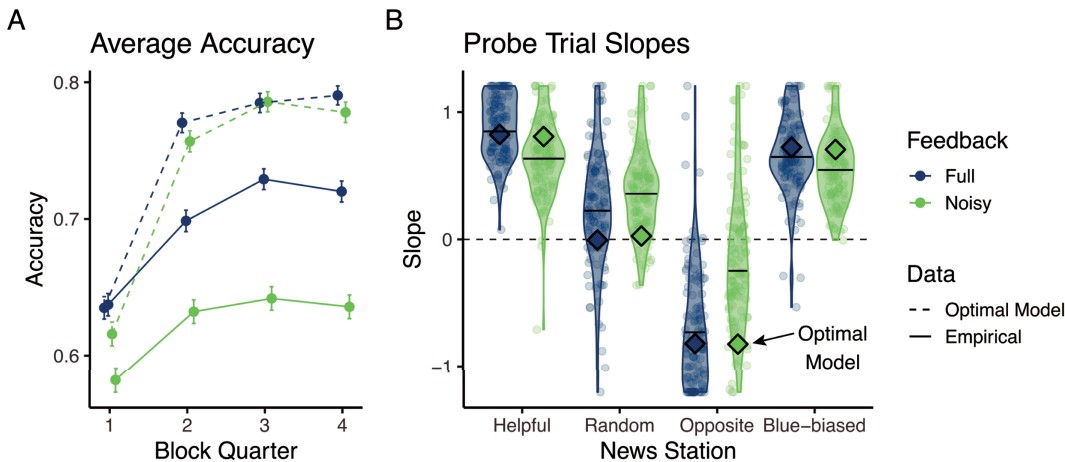

**Fig 9. Full feedback version shows better empirical performance.** (A) Average accuracy split by block quarter and experiment. We show both the average accuracy from the data (continuous line) and the average accuracy achieved by the optimal parameter-less model (dashed lines) operating with uniform priors. (B) Probe trials slopes by condition and news source. Mean slope values for the optimal model highlighted with framed diamonds.

difference increased over the course of the block (Fig 9A). This was supported by a significant trial number × experiment condition effect ($\chi^2(1) = 19.422, p < 0.0001$).

We also checked how far this performance diverged from the optimal model. To do so, we also ran the full normative models without fitting any parameters and operating with flat priors over the initial source probabilities ($b_{t=0,I} = g_{t=0,I} = Beta(1,1)$). We overlay these performances in Fig 9, panel A. This highlights that the optimal model significantly outperformed participants in both feedback conditions (significant trial number × optimal model effect, $\chi^2(1) = 184.17, p < 0.0001$). We also note how the performance differences between the two feedback conditions were small for the optimal model, showing the theoretical inferential power of an optimal semi-supervised model. Indeed, the optimal model licenses powerful inferences in both cases and asymptotes to high performance after the second quarter. In contrast, empirical data showed how participants struggled more with the noisy feedback condition.

Participants' probe trial responses were also closer to the optimal response in the full feedback condition (Fig 9B): Their average slope was higher in response to the helpful source and more negative in response to the opposite source (Tukey's tests comparing the conditions, $p < 0.0001$). The slope did not differ between the conditions for the blue-biased source (Tukey's test $p = 0.40$). We again overlay the optimal model's slopes in panel B which shows how participants' average slopes were close to the optimal model in the full feedback condition. We note how the comparably worse performance in the learning trials (panel A) is due to the much higher variability of participants' slopes (for example some participants positively integrating the random, or ignoring the opposite source).

## 2.7. Questionnaire analyses

### 2.7.1. Post-block trust and improvement ratings.
How might participants have assessed the news sources' qualities more broadly? To answer this, in both experiments, we asked participants two questions at the end of each block: One relating to how much they trusted the source and one relating to how much they thought the source improved their decisions.

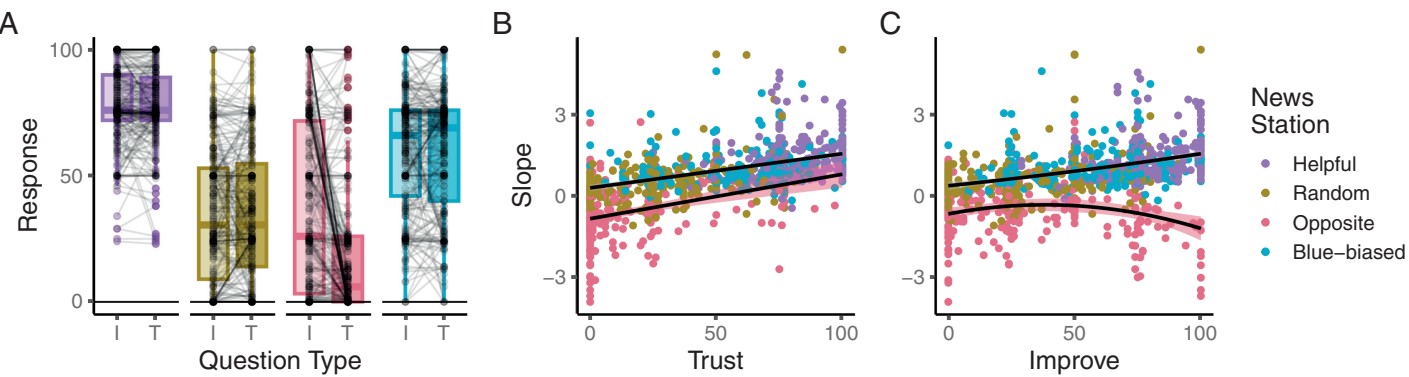

**Fig 10. Participants rate trust in sources and how they improve their decisions.** (A) Raw responses to post-block trust and improvement questions by source. (B, C) Relationships between second block half psychometric slopes and trust and improvement ratings. Regression lines fit jointly to helpful, random, and blue-biased sources, and separately to the opposite source.

Specifically, participants responded on a continuous scale ranging from "Strongly Agree" to "Strongly Disagree" (via intermediate options) to the following statements: (1) "I trust this news station" and (2) "This news station improves my decisions".

Analysis of participant responses revealed key distinctions between the sources (see Fig 10A): Throughout, participants rated the helpful source highest on both trust and improvement questions (Tukey's tests against all other sources, $p < 0.0001$). The blue-biased source was similarly rated high, albeit lower than the helpful source in both trust and improvement questions. The random source in turn was rated as neither trustworthy nor as particularly improving decisions and had lower trust and improvement ratings than the helpful and blue-biased sources (Tukey's tests $p < 0.00001$). A more curious pattern emerged in response to the opposite source: While it had the lowest average trust ratings (Tukey's tests against all other conditions $p's < 0.0001$), some participants indeed rated it highly for improving their decisions, in line with its general usefulness. Indeed, while all other sources had equivalent trust and improvement ratings (Tukey's tests $p's > 0.99$), the improvement rating of the opposite source was significantly higher than its trust rating ($p < 0.00001$), and as high as the improvement rating of the random source ($p = 0.9970$)

How did these responses relate to participants' behavior on the task? To address this, we correlated participants' psychometric slopes in the second block half to their trust and improvement ratings. We did so jointly for the helpful, random, and blue-biased source, and separately for the opposite source, for which we observed the trust-improvement distinction. The relationship between participants' trust ratings and their response slopes was best captured by a simple linear regression, highlighting that the more participants integrated the sources' information, the more they also rated it as trustworthy (Fig 10B, for regression model see Table D in S1 Text).

This pattern was more complex for the improvement ratings: For the helpful, random, and blue-biased source, the relationship between slopes and ratings was still best captured by a simple linear model. However, for the opposite source, this relationship followed a linear quadratic function: This meant that while some participants who realised they had to invert the source's information did not reflect this in their improvement ratings (negative slope, low improvement rating), other participants who had a similar realisation (negative slope) did reflect this in high improvement ratings.

**2.7.2. Trust and media consumption patterns.** How might behavior in our task relate to more real-world behavior and attitudes? To assess this, we administered a number of questionnaires relating to general interpersonal trust as well as trust in media in both experiments. Additionally, we investigated participants' media consumption patterns by asking them about their information sources and the frequency with which they engaged with these sources.

For a more principled approach to these questionnaires, we first performed an exploratory factor analysis on the entire set of 50 questions. This revealed a three factor structure: A first factor captured media consumption and trust, with consumption of (mainstream and social) media and trust in this media loading highly on this factor. A second factor represented both participants' levels of personal mistrust and credulity with respective questionnaire sub-scales displaying corresponding loadings. Interestingly, this factor also had weak negative loadings for the consumption of classical media and weak positive loadings for the consumption of social media. A final factor represented more general levels of interpersonal trust with questions around epistemic trust loading positively and questions around mistrust loading negatively (See S1 Text Text for more details).

We had no particular set of hypotheses regarding the relationships between these questionnaires and our task. We thus conducted a number of exploratory regression analyses linking the two. These analyses revealed no significant relationship between the factors extracted above and different behavioral components such as game scores, psychometric, or model parameters.

## 3. Discussion

We investigated people's ability to learn different statistical regularities of information providers. We framed this as a doubly Bayesian learning problem where agents learn about both a ground truth and the trustworthiness of a source, and developed a corresponding model. We showed how people were generally able to distinguish different kinds of information sources and respond to them adaptively. Participants integrated trustworthy information correctly. They also tended to learn to ignore useless information and invert information from a reliably incorrect source. Finally, they—albeit with limitations we discuss below—were able to uncover biases in news sources.

We conducted two experiments, in which participants received either full or noisy feedback. Across these tasks, we showed how participants followed the learning trajectories of a parameterized Bayesian model. Participants showed better performance in the task version with full feedback, in line with the more straightforward nature of the learning process involved. Our models described participants' learning patterns well. This is in line with previous conceptualizations of Bayesian learning about the trustworthiness of information sources, for example theoretically proposed in similar forms by [5] and [6] and empirically shown by [48] and [8].

Two behavioral signatures that differed from a purely optimal learning model emerged in both experiments. First, only few participants were able to learn the bias of the blue-biased source. This source is indeed the hardest to learn - requiring the tracking of two higher-order probabilities instead of one. Whether and how we deal with biased information is an ongoing question. For example, in a field experiment people adaptively took news bias into account for more accurate judgements [36], but an experimental study highlights errors in the way people seek information from biased sources [34].

Second, on average, participants tended to start out believing that our sources were helpful. This led them initially not to ignore the random source, or invert the opposite source. While such behavior is suboptimal in this specific context, a general bias holding information

providers to be useful may be generally adaptive. For example, most (social) media information we encounter tends to be true [49,50], and people are sensitive to the general distribution of misinformation in an environment [51]. Indeed, always starting to learn from scratch is wasteful, and holding sources to be generally useful would be a healthy and adaptive inductive bias in most non-adverse social environments [2,52]. We found a prior leaning more towards a "helpful" source in the more difficult task version with noisy feedback, indicating that participants might have relied more on the prior when learning was harder.

Our fit parameters allowed us to characterize some of these behaviors, along with providing other signatures: Participants used helpful priors, an inference also supported by model-agnostic analyses. In the full feedback condition, our model also allowed us to characterize participants' learning as being neither particularly hopeful nor unforgiving. The latter is interesting given that other investigations have found optimistic and pessimistic biases in Bayesian and reinforcement learning [53,54]. Particularly when it comes to interpersonal relations, trust is however often purported to be hard to build and easy to lose [55–57]. Similar effects have been shown with regard to news sources in more naturalistic experiments [58]. The fact that we do not find such a bias here might hint at this being not a general bias but rather a function of actors in a social environment.

We note that the purpose of our model fitting was to show general alignment between Bayesian-like learning patterns and participant behavior and not to make strong statements about a particular class of model or parameters. Indeed, we rely on a medium to high number of parameters that are partially intertwined. Future modelling will have to disentangle these factors. For example, resource-rational accounts might explain why only some participants learnt the bias [59]. To pin down specific effects such as the learning biases further, targeted experiments will also be necessary, for example, ones that might change the quality of a source mid-way through a block [33]. This will allow for more robust modelling conclusions [60].

Our post-block questionnaires showed that participants were able to explicitly express their trust in the sources and how strongly they believed that the sources improved their decision-making. We showed how this related to behavior in our task. This points to the validity of our paradigm. The fact that these ratings also dissociate for the opposite source hints that people don't purely understand trust to be a merely utilitarian construct [61]. The ratings also demonstrate how people express divergent trust towards sources we may consider "bullshit" and those that lie [28]. It would be interesting to consider how trust and utility trade off: For example, would we rather (pay to) hear from a useful but untrustworthy liar or a less useful but trustworthy source?

Our broader questionnaire measures did not correlate with individual differences in our task. There are several lenses through which to view this finding: First, our tightly controlled task came at the cost of external validity, and the statistical learning task we presented participants with was necessarily abstract. More realistic extensions of our paradigm may present participants with actual news sources like newspapers and with more realistic stimuli like news headlines. This would naturally be accompanied by a drop in control and increase the difficulty of quantifying the individual likelihoods associated with each item.

Second, our pure Bayesian ability to learn the trustworthiness of information sources might only be a marginal factor driver behind our real-world attitudes and media choices. Instead, forces like motivated reasoning [15,46] might explain more variance. Additionally, we often do not just learn about news sources through direct interaction like we described here but also hear about the reliability of sources from other sources. This can create triple (or indeed quadruple) learning problems. However, it is generally important to note that we

did not set out this study to focus on individual differences, and larger samples are usually necessary to pin down individual differences in this field [62–65].

The role that sources play in helping people identify misinformation is contested. Research into boosting people to be more robust towards misinformation indeed tries to highlight the source [29,66]. Indeed, people tend to generally have a good assay of the trustworthiness of news providers, at least in a US context [25], and displaying explicit trustworthiness rating may lower their propensity to share fake news [67]. However, other research [13] shows that people can pay little regard to the sources of misinformation, making them more vulnerable to being misled. One prominent mechanism for fostering this vulnerability is the tendency of individuals to conform with majority opinion [68]. Even in the simplified setting of our experiment, participants may experience the need to align their beliefs with the majority opinion (represented, e.g., by the council of experts), potentially putting lower weight on their own assessment of the trustworthiness of sources. This bias is reflected in the priors observed in the experiment, which suggest an inherent assumption that the majority is right. This bias is often [69–71], though not always, useful – a fact that malign actors can exploit.

The inferences our participants and models draw about news providers can be understood as a basic form of theory of mind [72–75]: In essence, participants reason and learn about how the sources produce evidence from the ground truth by inverting their model via Bayesian inference. This type of inverse reinforcement learning has been posited as a key computational substrate for shallow forms of Theory of Mind (ToM) [76]. More complex forms of theory of mind can also take into account the beliefs and intentions of other agents, and those other agents' recursive reasoning about the participants themselves [77]. In broader terms, the ability to deal with such intentions plays a significant role in the consumption and interpretation of media and in psychopathology. When we suspect a source might profit from misleading us, we become more skeptical towards it, and its messages might even backfire [31] or be ignored as cheap talk [78]. Computationally, this can be modelled as a recursive theory of mind, where a sender tries to hack the inference process of a receiver and where this receiver might try to infer about such a hacking scheme and defend against it [30,32]. Over- or under-interpreting other agents within this cognitive hierarchy can be understood as both credulity and paranoia, important for different aspects of mental disorders [79,80]. It would be interesting to consider the effects of such recursive modeling in our task.

A defining feature of the modern information sphere is abundant choice. Particularly in the political realm, we can rather freely decide between news sources [81,82]. Future iterations might take into account this information-seeking perspective [83–86], adding key computational challenges but potentially letting us observe interesting behavioral patterns (see [87] for an empirical investigation of this using a perceptual task). Specifically, agents that have free reign over their sources need to decide when they still need to explore different sources and when they can begin exploiting a source's knowledge [88]. Such exploration-exploitation dynamics also open the door for a number of path dependencies that might make participants disengage with potentially mistakenly erroneous news sources quickly, akin to the hot stove effect [89–91].

In summary, our results have optimistic and pessimistic implications: On the optimistic side, our subjects were generally very successful at their dual Bayesian learning task, showing that the basic components of accurate source assessment and integration have not been dulled. However, that our subjects struggled to learn about the bias of a skewed source even in our rarefied abstract conditions leads to pessimism that we will be able to avoid blandishments exploiting motivated reasoning, and worse. Work on prevention and cure is pressing.

## Supporting information

**S1 Text. Supplementary text containing additional information and analyses.**
(PDF)

## Author contributions

**Conceptualization:** Lion Schulz, Yannick Streicher, Eric Schulz, Rahul Bhui, Peter Dayan.

**Data curation:** Lion Schulz, Yannick Streicher.

**Formal analysis:** Lion Schulz, Yannick Streicher, Eric Schulz, Peter Dayan.

**Funding acquisition:** Peter Dayan.

**Investigation:** Lion Schulz, Yannick Streicher, Eric Schulz, Rahul Bhui, Peter Dayan.

**Methodology:** Lion Schulz, Yannick Streicher, Eric Schulz, Rahul Bhui, Peter Dayan.

**Project administration:** Lion Schulz, Peter Dayan.

**Resources:** Rahul Bhui, Peter Dayan.

**Software:** Lion Schulz, Yannick Streicher.

**Supervision:** Lion Schulz, Eric Schulz, Rahul Bhui, Peter Dayan.

**Validation:** Yannick Streicher, Peter Dayan, Lion Schulz.

**Visualization:** Lion Schulz, Yannick Streicher.

**Writing – original draft:** Lion Schulz, Yannick Streicher, Peter Dayan.

**Writing – review & editing:** Lion Schulz, Yannick Streicher, Eric Schulz, Rahul Bhui, Peter Dayan.

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
