## [Decision Letter · Decision Letter 0]

7 Feb 2024

Dear Mr. Schulz,

Thank you very much for submitting your manuscript "Mechanisms of Mistrust: A Bayesian Account of Misinformation Learning" for consideration at PLOS Computational Biology.

As with all papers reviewed by the journal, your manuscript was reviewed by members of the editorial board and by several independent reviewers. In light of the reviews (below this email), we would like to invite the resubmission of a significantly-revised version that takes into account the reviewers' comments.

We cannot make any decision about publication until we have seen the revised manuscript and your response to the reviewers' comments. Your revised manuscript is also likely to be sent to reviewers for further evaluation.

Sincerely,

Christoph Mathys

Academic Editor

PLOS Computational Biology

Thomas Serre

Section Editor

PLOS Computational Biology

Reviewer's Responses to Questions

**Comments to the Authors:**

Reviewer #1: Schulz et al. addressed the mechanisms of trust/mistrust using a Bayesian learning framework and carefully designed experiments. Across two studies, they examined how participants use information from (a) helpful, (b) random, (c) opposite, and (d) biased sources, after receiving full (Stidu 1) and partial (Study 2) feedback. The analysis followed the psychometrics function as motivated by the literature. The results mostly followed the theoretical predictions, and there are interesting differences between full vs partial feedback.

This manuscript is very clear and well-written, and the topic is important and timely, especially nowadays information/misinformation on social media plays a key role in everyday life. I only have a handful of clarification questions that hopefully will improve the paper.

(1) What is the meaning of the three small triangles in Fig1B? Does it indicate a news station? It’d be great to make it explicit (also for Fig 5).

(2) In line 160, it would be helpful to define the meaning of Y already here, rather than later (line 168).

(3) line 175 “the agent combines its prior […] to form an updated belief about the two parameters […]” It seems that the inferred two parameters are b and g; but following the reasoning of this paragraph, the two parameters might refer to the slope and intercept in the psychometric function? Please clarify.

(4) In general, what are the parameters in the model? Indeed more details are provided in the SI, but it would be very helpful for readers to clearly see the notions and meanings of all free parameters, eg, by including a table.

(5) I am not entirely convinced by the twice data split. Isn’t it the case that blocks 1-2 are the first half of the experiment, and blocks 3-4 are the second half of the experiment? So technically, the data is analyzed twice, just by different split criteria? This seems arbitrary, and knowing the data from 4 blocks, one could already infer the results with the half-split. I would rather leave out the half-split results.

(6) For the model belief dynamics (Fig 4E, 8E), it would be helpful to actually plot the beliefs for blue and green separately, rather than the average. This matters because the design is framed with respect to the blue information, particularly for the biased information source. Hence it’d be great to see the beliefs for green and blue separately.

(7) I was rather intrigued by the differences in the priors between the two studies. Indeed, in both cases, the prior is more positive (above 0.5), but clearly, a large proportion of participants’ prior in Study 1 is around 0.5, whereas most participants’ prior in Study 2 is about 0.75. It would be useful to provide some interpretation (around Line 561).

(8) Fig 10; in addition to the relationship between Trust/Improve and psychometric parameters, I wonder if Trust/Improve is associated with some parameters from the Bayesian learning model?

Reviewer #2: In their work ‘Mechanisms of Mistrust: A Bayesian Account of Misinformation Learning’ Schulz et al. study the way humans learn about the accuracy of information sources. They used a novel experimental paradigm and a computational Bayesian learning model to uncover this phenomenon. In two experiments, they presented a panel of opinions as a source of information, which could be accurate, non-informative, biased, or wrong (or misleading). They found that participants learn the relation between the information source and the real state of the world, as reflected in their estimations. When the outcome was ambiguous, and participants had to infer both the state of the world and the source accuracy, participants found it harder to estimate the bias and information content of the information source. Their work shows that participants prior about source accuracy, coupled with uncertain outcome, greatly affect estimation of source accuracy.

This work presents compelling computational framework and experimental design and provides important evidence concerning the way people tackle uncertainty in social learning. I personally found the results from the random condition to be very interesting, as people tend to follow them as if they contain signal and fail to distinguish between noisy and coherent sources (especially when outcome is ambiguous).

I think that the approach and methods are appropriate, and the paper is written in a clear way.

I have a number of technical comments below, and one theoretical comment.

Theoretical comment:

The authors frame their study as a way to evaluate trustworthiness of information sources, such as media outlets. However, unlike studies that directly use headlines from newspapers, social media or tv, or prepare stimuli that is very similar to these, they use a panel of opinions to convey an opinion. I am not concerned about this reduction, but I do think that this display brings some biases that the authors do not discuss. There is literature about majority opinion and its epistemic value and tendency to conform. The conformity works of Asch look at the effects of panel opinion, and many followers examined how different majorities affect individual decisions to comply with the group. This is even in cases where the group is known to be wrong, showing some motivation or bias to conform to the majority that may also be relevant to the current experimental design. Other works examine the epistemic value of majority, going back to Condorcet’s jury theorem, and more recent works on informational cascades, economics research ( Shmuel Nitzan’s works for example), and in moral judgment (Lindström et al. (2018). common is moral). I think that this literature may be useful in framing and interpreting the results, and also to highlight the relevance of the experimental paradigm. Note that this may have some implications for your model, as these approaches assume that each panel member is an independent observation of the noisy world.

In addition, the current discussion linking the learning model to theory of mind seems a little forced. For example, it is not clear what one makes of the opposite group – it is not likely to face a source that faithfully tells you the opposite of the real state. What kind of theory of mind is at work here? In the sense that inferring the underlying distribution is a model of the source I agree that this is indeed the case, but I find that this link on its own is not very fruitful for understanding the way people infer about sources of information in its current state.

Technical point:

1. Was confidence ratings distribution similar between conditions? The random case seems to be most vulnerable to participants giving mostly low confidence ratings. Did this affect model fit/results?

2. Line 260 – model fitting – some link to the model fitting procedure could be useful, as well as summary statistics about the parameter estimations (mean and range, for example) and indication of goodness of fit.

3. Line 276 – binned analysis – I wonder whether using mixed effect logistic regression, with individuals’ bias and slope as random variables, may not be a better way to study both individual level and group effects, instead of pulling ratings across participants and addressing individual slopes in two separate analyses.

4. Fig 9. It was very hard to see the empirical mean in panel B – lines are very similar to background.

5. Line 490 – slope and trust/intercept – there seem to be missing statistics in the text. Also, is the linear relation significant, or a mean value better describe these relations?

6. Line 550 - adaptively is written twice.

Reviewer #3: Summary:

This paper presents a new learning paradigm in which participants must infer the underlying state (blue/green) based on a signal as well as the reliability of the source of the signal (the news agency). Two experiments are reported. In the first experiment, participants receive unambiguous feedback on each trial about the true value (blue/green). In the second experiment, they receive ambiguous feedback in the form of several unbiased samples from an "independent council". Four kinds of news stations are tested: helpful, randomm, opposite, and blue-biased. Participant behavior is analyzed with a Bayesian model that also includes additional parameters (totling 9 parameters). The main result is that people in the first experiment are better at learning the news station mapping than in the second experiment. Additionally, people appear to perform much worse than optimal Bayesian.

Comments:

The paper tackles an interesting question and introduces a simple paradigm for understanding how people jointly reason about content and reliability of an information source. However, the main results seemed unsurprising, and even though there are interesting model-based analyses of the data, the paper would be improved if they were used in more of a way to advance an argument or investigate the underlying mechanisms at play. Some more specific points:

1. When introducing the model in the main text, the authors give very little intuition for what the model is supposed to capture other than that its Bayesian plus some other biases needed to fit the data. Why are these biases needed to capture the data? What happens if you fit the models without those biases? Some discussion of the qualitative features of the models with/without the biases is needed since one of the main takeaways from both experiments is that people *are not* Bayesian, even though the paper introduces the Bayesian solution as the working hypothesis for what people would be doing in the task.

2. Building on the previous point, the authors should fit the data with different lesioned versions of the model and report the relative fits. That will give us a better sense of the work that each component of the model is doing and aid in interpreting the data.

3. Its possible that people are not just Bayesian plus some biases, but using relatively simple heuristics or learning rules to solve the task. It would be good if the authors also tested some simple decision procedures used in the multi-armed bandit literature (since this task could be seen as a bandit with a slightly more abstract structure than usual). For example, could win-stay/lose-shift over trust/not-trust a news source be a strategy? What about a simple (non-Bayesian) reinforcement learning model like Rescorla-Wagner? One of the virtues of using such a simple task to study how people reason about potential sources of misinformation is that a more comprehensive model comparision is possible!

4. It wasn't clear to me from the main text that people were incentivized to learn about the truth.

5. "To answer this, we next analyzed participants’ learning trajectories over the first 28 trials of a block." - does this just mean you're not analyzing the probe trials? It would be clearer if this was stated directly maybe by referring to these as the "main trials".

6. "demarcated probe trials" - were these demarcated to the participants?

7. I couldn't find an explicit statement about how many blocks of trials were in each experiment.

8. " Participants played with each of the sources for a total of 28 trials. " - "played" was a peculiar choice of word and I wasn't sure if it just meant they received 28 trials for that source or interacted with the interface in a more extended way.

**Have the authors made all data and (if applicable) computational code underlying the findings in their manuscript fully available?**

Reviewer #1: **No: **

Reviewer #2: None

Reviewer #3: **No: **The data/code are not currently available but the authors state they will provide it upon publication.

PLOS authors have the option to publish the peer review history of their article (what does this mean?). If published, this will include your full peer review and any attached files.

Reviewer #1: No

Reviewer #2: No

Reviewer #3: No
---

## [Decision Letter · Decision Letter 1]

21 Jan 2025

Dear Mr. Schulz,

We are pleased to inform you that your manuscript 'Mechanisms of Mistrust: A Bayesian Account of Misinformation Learning' has been provisionally accepted for publication in PLOS Computational Biology.

Best regards,

Christoph Mathys

Academic Editor

PLOS Computational Biology

Thomas Serre

Section Editor

PLOS Computational Biology

Reviewer's Responses to Questions

**Comments to the Authors:**

Reviewer #1: The authors have done a very good job at address my original questions.

I would only suggest the authors include the current Table S2-3 regarding transparency of model parameters to the main text.

I look forward to seeing this out soon.

Reviewer #2: The authors addressed all my comments adequately. I don't have any further comments.

**Have the authors made all data and (if applicable) computational code underlying the findings in their manuscript fully available?**

Reviewer #1: **No: **the authors declared that they will make them publicly available when the paper is accepted/published.

Reviewer #2: **No: **The code and data are not yet available - they state that they will make them available upon publication.

PLOS authors have the option to publish the peer review history of their article (what does this mean?). If published, this will include your full peer review and any attached files.

Reviewer #1: No

Reviewer #2: No

---

## [Editor Report · Acceptance letter]

PCOMPBIOL-D-23-01750R1

Mechanisms of Mistrust: A Bayesian Account of Misinformation Learning

Dear Dr Schulz,

I am pleased to inform you that your manuscript has been formally accepted for publication in PLOS Computational Biology. Your manuscript is now with our production department and you will be notified of the publication date in due course.

With kind regards,

Anita Estes
